# Latest Findings on Minimally Invasive Anatomical Liver Resection

**DOI:** 10.3390/cancers15082218

**Published:** 2023-04-09

**Authors:** Yoshiki Fujiyama, Taiga Wakabayashi, Kohei Mishima, Malek A. Al-Omari, Marco Colella, Go Wakabayashi

**Affiliations:** 1Department of Surgery, Ageo Central General Hospital, Saitama 362-8588, Japanmalekamo86@yahoo.com (M.A.A.-O.); mrccolella@gmail.com (M.C.); gowaka@ach.or.jp (G.W.); 2Center for Advanced Treatment of Hepatobiliary and Pancreatic Diseases, Ageo Central General Hospital, Saitama 362-8588, Japan; 3Institute for Research against Digestive Cancer (IRCAD), 67091 Strasbourg, France

**Keywords:** MIALR, Tokyo 2020 terminology, ICG negative staining

## Abstract

**Simple Summary:**

The safety of minimally invasive anatomical liver resection is a major concern for hepatobiliary surgeons. The Precision Anatomy for Minimally Invasive Hepato-Biliary-Pancreatic Surgery Expert Consensus Meeting was held in 2021. In this meeting, the importance of intraoperative staining of the dominant portal venous region was confirmed, with indocyanine green playing a central role. This article describes the latest findings on minimally invasive laparoscopic anatomical liver resection using the indocyanine green negative staining technique.

**Abstract:**

Minimally invasive liver resection (MILR) is being widely utilized owing to recent advancements in laparoscopic and robot-assisted surgery. There are two main types of liver resection: anatomical (minimally invasive anatomical liver resection (MIALR)) and nonanatomical. MIALR is defined as a minimally invasive liver resection along the respective portal territory. Optimization of the safety and precision of MIALR is the next challenge for hepatobiliary surgeons, and intraoperative indocyanine green (ICG) staining is considered to be of considerable importance in this field. In this article, we present the latest findings on MIALR and laparoscopic anatomical liver resection using ICG at our hospital.

## 1. Introduction

With recent developments in minimally invasive surgeries, laparoscopic and robot-assisted procedures have become increasingly popular for hepatobiliary surgery [1,2,3,4,5]. Two international consensus conferences [6,7] and one international guideline conference [8] have been held to promote the safe use of minimally invasive liver resection (MILR). Safe and minimally invasive anatomical liver resection (MIALR) is the next keystone in hepatobiliary surgery [9,10,11,12]. To address the safety of MIALR, the Precision Anatomy for Minimally Invasive Hepato-Biliary-Pancreatic (HBP) Surgery Expert Consensus Meeting (PAM-consensus meeting) was held in Japan in 2021 [13,14]. Based on this meeting, the Tokyo 2020 Terminology of Liver Anatomy and Resections (Tokyo 2020 terminology) was recommended as an important guideline [15]. The Tokyo 2020 terminology defines anatomical liver resection more clearly, updating the Brisbane 2000 terminology of liver anatomy and resections [16,17,18]. The PAM consensus meeting and the Tokyo 2020 terminology addressed the importance of regional staining for safe MIALR. We believe that indocyanine green (ICG) plays a central role in MIALR, and that liver resection along the intersegmental plane with ICG negative staining is an acceptable basic technique for MIALR. This paper describes the latest findings on MIALR and our MIALR technique using ICG green-negative staining.

## 2. Latest Findings on the MIALR–PAM Consensus Meeting and Tokyo 2020 Terminology

MIALR has gradually become widely available worldwide following two international consensus conferences and one international guideline conference. However, the definition of MIALR and its associated surgical procedures have not been well formulated. An expert consensus meeting on the Precision Anatomy for Minimally Invasive HBP Surgery (the PAM consensus meeting) was held during the 32nd Annual Meeting of the Hepatobiliary and Pancreatic Surgery Society in February 2021. Hepatobiliary surgeons worldwide shared their opinions regarding the definition and safe surgical procedures for MIALR [9,19,20,21]. The main points of the PAM consensus meeting were published as the Tokyo 2020 Terminology of Liver Anatomy and Resections (Tokyo 2020 terminology). The Tokyo 2020 terminology is an important milestone in the field of liver surgery. The key points are listed below.

Key point 1: MIALR is defined as the complete resection of the liver parenchyma within the region of the respective portal vein.

Key point 2: Segmentectomy is defined as the complete resection of the third portal vein branch of the Couinaud classification. Subsegmentectomy is defined as the resection of the area beyond the third portal vein branch of the Couinaud classification, with each area defined as a “cone unit.”

Key point 3: Two approaches to accessing the Glisson’s sheath have been described: the Glissonean approach (GA), which secures Glisson’s sheath in one piece, and the hilar approach (HA), which separates the artery, portal vein, and bile ducts individually. In MIALR, the GA is more suitable than the HA, particularly when approaching peripherally from secondary Glissonean branches.

Key point 4: The intersegmental/sectional plane (IP) is the boundary of each portal-dominant territory, and the vein passing through the IP is defined as the intersegmental/sectional vein (IV). Preoperative three-dimensional (3D) simulations are useful for visualizing the IPs and IVs. The dissection of the liver parenchyma along the correct IP is crucial for a precise anatomical liver resection.

Key point 5: To visualize the IPs in MIALR, negative staining with intravenous contrast after blocking the target Glisson or positive staining via direct injection into the portal vein is recommended.

## 3. Laparoscopic Anatomic Liver Resection at Ageo Central General Hospital (ACGH)

As of April 2016, all laparoscopic liver resections without biliary or vascular reconstruction are covered by health insurance in Japan. Thanks to this coverage, we performed 213 laparoscopic anatomical liver resections as of March 2022 (Table 1) [22,23,24,25]. In some cases, subsegmentectomy targeting the periphery beyond the third branch of Glisson’s sheath was performed to achieve both anatomical resection and sparing of the liver parenchyma. All liver resections without biliary or vascular reconstruction were considered as indications for MIALR. Regarding the indications for anatomical resection, we performed anatomical resection for hepatocellular carcinoma, except in cases with a peripherally located tumor. For diseases other than hepatocellular carcinoma, we chose anatomical resection when the carcinoma-bearing Glisson was more central than the tertiary branch. This is because we believe that anatomical resection is a more physiological procedure that follows hepatic blood flow. The indications for the 213 patients and short-term outcomes are shown in Table 1 and Table 2, respectively. As shown in the tables, our MIALR procedure was characterized by the anatomical resection of metastatic tumors (Table 1) and a large number of segmentectomy/subsegmentectomy cases (Table 2).

The main points of our laparoscopic anatomical liver resections are as follows: (1) preoperative 3D simulation imaging based on the cone unit theory and intraoperative 3D monitoring, and (2) an intraoperative Glissonean approach to secure the target Glisson and dissection of the liver parenchyma along the intersegmental plane using indocyanine green (ICG) negative staining. The details of our surgical technique are presented below.

### 3.1. Preoperative 3D Simulation Imaging and Intraoperative 3D Monitor

In all cases, 3D simulation images were constructed based on the preoperative dynamic computed tomography (CT) data (Zaiostation2, Ziosoft Co., Tokyo, Japan). Deciding which Glisson should be divided and which should be preserved is crucial to this technique. Based on the division/preservation, a hepatic dissection plane can be constructed, and a surgical outline can be created (Figure 1a–c). Intraoperatively, a 3D monitor (Atrena, Amin Co., Tokyo, Japan) was routinely used to project preoperative 3D simulation images. The advantage of using the Atrena intraoperatively is that the actual Glissonean branching can be compared with the simulated 3D images. In addition, 3D information regarding the distance or direction from the hepatic hilum can be used to accurately identify tumor-bearing Glissonean pedicles. The Atrena can be operated on a touch panel in a clean surgical field; therefore, Glissonean branching can be assessed and discussed in real time (Figure 1d). In our study, the error rate between the predicted liver resection volume calculated using the preoperative simulation and the actual resection volume was within 10% in more than 80% of all cases. Thus, we believe that creating a preoperative 3D simulation is useful for a safe and precise laparoscopic liver resection.

### 3.2. Liver Parenchyma Dissection with the Glissonean Approach and ICG Negative Staining

In our laparoscopic anatomical liver resections, the Glissonean approach was used to identify carcinoma-bearing Glissonean tumors by sequentially following them from the primary Glissonean branch to the periphery [26,27]. The target Glissonean pedicle was blocked with a bulldog clamp and ICG staining was performed using a basic technique [28,29]. The standard intravenous dose administration of ICG is 0.5 mg/body. When observed with an ICG camera (1688 AIM 4 K camera system, Stryker), a clear ICG demarcation between the fluorescent (preserved liver) and nonfluorescent (resected liver) areas appeared on the liver surface. Importantly, this ICG demarcation line was identified not only on the liver surface but also within the liver parenchyma, which is the dissecting plane of an anatomical liver resection IP. We believe that the risk of postoperative bile leakage is low if anatomical hepatic resection is performed along the correct IP because there are no transverse Glissonean sheaths across the IPs. Figure 2, Figure 3 and Figure 4 show intraoperative images of left hepatectomy, posterior sectionectomy, and S3 subsegmentectomy.

Robot-assisted hepatectomy has been covered by health insurance in Japan since April 2022. We performed robot-assisted hepatectomy using the same concepts as laparoscopic hepatectomy, namely, the Glissonean approach and ICG negative staining. We have performed 21 cases of robot-assisted liver resection as of March 2023. The multiarticulation unique to robotic surgery overcomes the limitations of the movement of surgical instruments restricted by laparoscopic ports. The articulation of robotic surgery facilitates the approach to vessels such as the Glisson and hepatic veins from any angle, which is a major advantage over laparoscopic surgery.

## 4. Discussion

One of the most significant technical differences between open and laparoscopic surgery is the angle of the surgical field of view; open surgery provides a cranial view, whereas laparoscopy provides a caudal view [30]. The hepatic-vein-guided approach (HVGA), a standard technique in open anatomical liver resections, is a vein-guided dissection of the liver parenchyma from the main hepatic vein to the periphery, which is a reasonable technique for open surgery in the cranial view. On the other hand, laparoscopic surgery, which provides a caudal view, is more suitable for the approach from the hepatic hilum as compared to open surgery. Therefore, the Glissonean approach (GA), which secures the Glissonean pedicles from the hepatic hilum, is a suitable surgical technique that takes advantage of laparoscopic surgery. Importantly, the HVGA and GA are not conflicting surgical approaches. The HVGA also plays an important role in MIALR by preventing disorientation during the liver parenchymal dissection, which is considered a pitfall of MIALR. Using the HVGA during MIALR has been advocated for by the PAM consensus/TOKYO 2020, and it is recommended that the tip of the device should be moved from the root to the periphery to prevent split bleeding [13]. Importantly, anatomical liver resection is not performed along the hepatic vein, but along the portal-vein-dominated area, based on Couinaud’s definition. Therefore, exposing the major hepatic vein is not in itself the goal of anatomical liver resection, but is one of the techniques for safe liver resections. In our MIALR using the Glissonean approach and ICG negative staining, the dissection plane of the liver cut surface was not always flat, and the major hepatic vein was often not fully exposed.

At the PAM consensus meeting, visualization of the portal-dominant region using ICG during MIALR was recommended. However, whether negative or positive staining (PS) is the better technique for MIALR is a topic for future studies. In 2008, Aoki et al. reported the usefulness of positive staining in open anatomical liver resection [31]. In 2021, Felli et al. published a review paper on positive or negative staining, concluding that further case reports needed to be accumulated under standardized surgical conditions, including the dose and injection speed of ICG or observation devices. From our experience, PS seems to have greater technical difficulty and instability, but this may be due to a technical bias. The indications for PS and NS may differ depending on the tumor site, type of liver resection, and other factors. Subsegmentectomy for segment5, far from the hepatic hilum, may be a good indication for PS, whereas segmentectomy for segment7—in which it is technically difficult to puncture the target Glissons from the body surface and close to the hepatic hilum—may be a good indication for NS. The superiority of PS over NS is an important topic for future research.

The validity of anatomical hepatectomy for HCC has been widely reported, but is currently controversial for CRLM [32,33,34]. As shown in the results, we often performed anatomical resections for CRLMs. Anatomical hepatectomy for CRLM may seem contrary to the concept of parenchyma-sparing surgeries. However, in some cases, we believe that liver parenchymal preservation and anatomical hepatectomy are compatible with the selection of Glissons after the third bifurcation of Couinaud, which is defined as a subsegmentectomy in the Tokyo 2020 terminology. Hepatic parenchymal dissection along the cone unit should theoretically have no Glisson’s transection and may be an ideal hepatic resection. We are in the process of accumulating cases of anatomic liver resection for CRLM for further reports.

There is a technical learning curve for liver resection using the Glissonean approach and ICG-NS. We believe that the key to safely overcoming the learning curve is the standardization of surgical procedures. When adopting these techniques for liver resection, standardization plays an important role in sharing surgical strategies with the surgical team. The main points of our standardized procedure are as follows: (1) Construct a preoperative 3D simulation image based on the concept of cone units. (2) Use an intraoperative 3D monitor. (3) Essentially, adapt the same patient position, port placement, and surgical instruments. (4) Encircle the responsible Glissonean pedicles using the GA. (5) Tape and clamp the target Glissonean pedicles. (6) Conduct intraoperative US with perflubutane to check if simulated ischemic boundaries can be observed in the liver parenchyma. (7) Carry out ICG NS (always 0.5 mg/body iv). (8) Perform liver parenchyma transection along the intersegmental plane. (9) Use antiadhesion agents in preparation for repeat hepatectomy.

The technological evolution of minimally invasive surgery has brought about a new phase in liver resection, namely, the widespread use of robotic-assisted hepatectomies. The articulating capabilities of robotic surgery are very useful for the dissection of dorsal aspects, such as the Glisson and hepatic veins. Moreover, the double-console system, which allows for a one-click changeover of surgeons from trainees to proctors, helps trainees to train in robotic surgery efficiently and safely. However, for hepatic parenchymal resection, we look forward to further improvements in the performance of ICG cameras and surgical devices for robotic-assisted hepatectomy. Currently, ICG cameras for robotic surgery do not have an overlay feature, which makes the background appear darker during anatomical liver resection. We believe that robotic-assisted anatomical liver resections with the GA and ICG-NS can be performed more safely as the scope of ICG and the devices in robotic surgery improve.

## 5. Conclusions

Intraoperative ICG staining plays a central role in MIALR. MIALR with the Glissonean approach and ICG negative staining, which is our standard technique, is a useful technique using ICG.

## Figures and Tables

**Figure 1 cancers-15-02218-f001:**
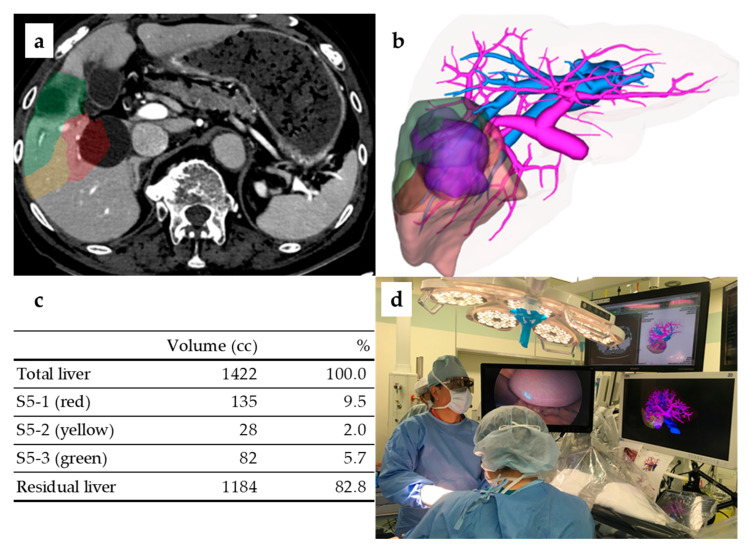
Preoperative 3D simulation (S5 segmentectomy) and intraoperative 3D monitor. (**a**) Segment5 in this case consists of three cone units. (**b**) 3D-constructed image. (**c**) Liver volume calculation per cone unit. (**d**) 3D glasses were used during surgery, and simulated images were projected on a 3D monitor (center).

**Figure 2 cancers-15-02218-f002:**
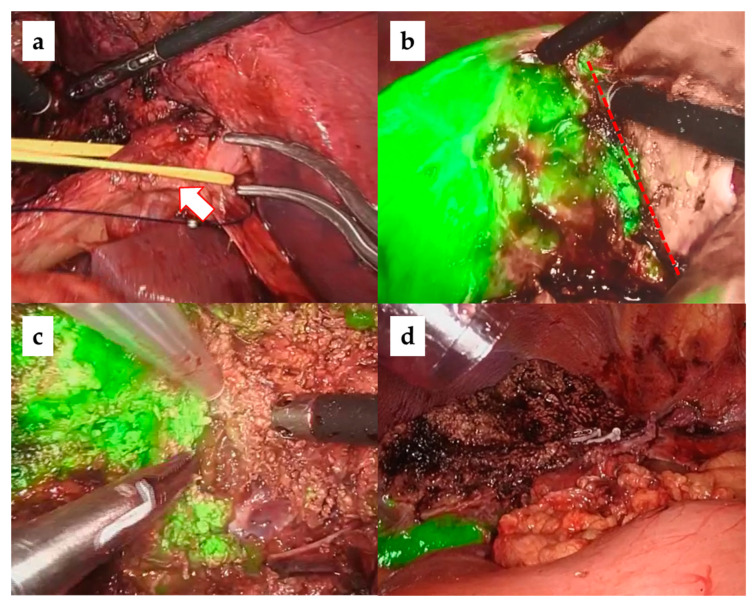
Laparoscopic left hepatectomy with ICG negative staining. (**a**) Securing and blocking the main trunk of the left branch of Glisson (arrow). (**b**) ICG demarcation line is observed in the center of the gallbladder bed (dotted line). (**c**) Liver parenchymal dissection along the intersegmental plane. (**d**) Liver cut surface.

**Figure 3 cancers-15-02218-f003:**
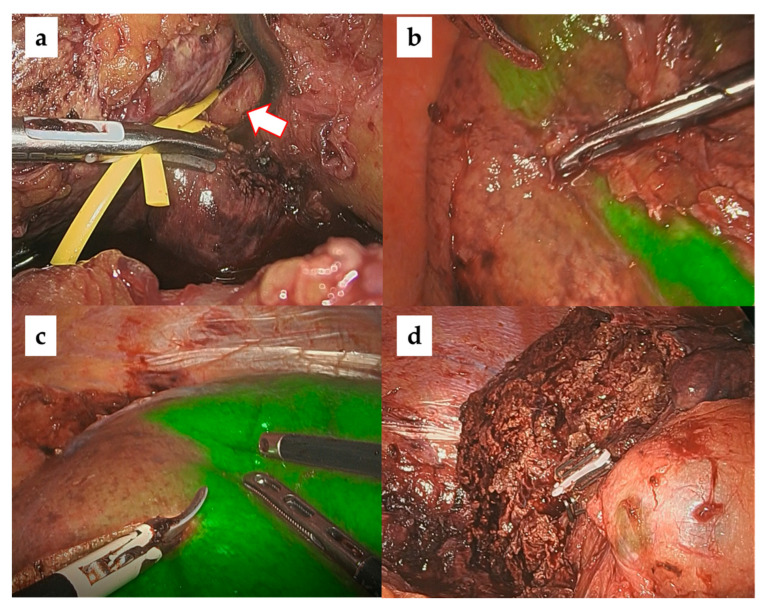
Laparoscopic right posterior sectionectomy with ICG negative staining. (**a**) Securing the main trunk of the right posterior branch of Glisson (arrow). (**b**) ICG demarcation line (dorsal). (**c**) ICG demarcation line (ventral). (**d**) Liver cut surface.

**Figure 4 cancers-15-02218-f004:**
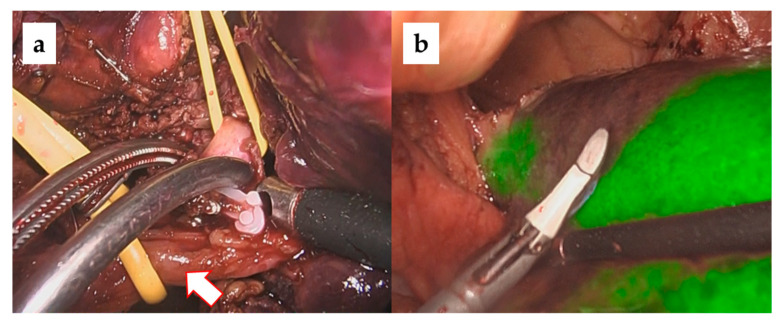
Laparoscopic subsegmentectomy (S3) with ICG negative staining (**a**) Securing and blocking the one of the branches of G3 (arrow: main trunk of G3). (**b**) ICG demarcation of subsegmentectomy of S3 (cone unit resection).

**Table 1 cancers-15-02218-t001:** Indications of 213 MIALR cases.

Disease	n (%)
Hepatocellular carcinoma	104 (48.4%)
Metastatic tumor	72 (33.8%)
Intrahepatic cholangiocarcinoma	16 (7.5%)
Benign liver disease	11 (5.2%)
Malignant lymphoma	3 (1.4%)
Others	7 (3.3%)

n = 213.

**Table 2 cancers-15-02218-t002:** Types of surgical procedure and surgical outcome.

	Case (%)	Procedure		Operation Time (min)	Blood Loss (mL)	Conversion to Open	Postoperative Complication (CD ≥ IIIa)	Postoperative Hospital Stay (Day)
Hr3	1 (0.5%)	Left trisectionectomy	1 (0.5%)	354	188	0 (0%)	1 (100%)	47
Hr2	40 (19.4%)	Left hepatectomy	20 (9.4%)	353 (216–540)	210 (15–2737)	0 (0%)	1 (5.0%)	8 (5–15)
		Right hepatectomy	14 (6.6%)	448 (305–798)	118 (10–925)	0 (0%)	1 (7.1%)	11.5 (6–53)
		Central bisectionectomy	6 (2.8%)	372 (281–542)	646 (80–1241)	0 (0%)	1 (16.7%)	13.5 (6–142)
Hr1	60 (27.8%)	Right anterior sectionectomy	20 (9.4%)	389 (214–552)	429 (47–1881)	0 (0%)	4 (20.0%)	11 (6–90)
		Right posterior sectionectomy	19 (8.9%)	405 (304–639)	456 (5–1523)	1 (5.3%)	0 (0%)	11 (5–21)
		Left medial sectionectomy	15 (7.0%)	331 (215–420)	190 (5–867)	0 (0%)	0 (0%)	9 (6–97)
		Left lateral sectionectomy	6 (2.8%)	267 (149–439)	293 (35–2367)	0 (0%)	1 (16.7%)	9 (5–15)
HrS	112 (54.4%)	Segmentectomy	88 (41.3%)	338 (110–850)	193 (10–5600)	0 (0%)	10 (11.4%)	9 (5–251)
HrSS		Subsegmentectomy	24 (11.3%)	335 (163–585)	173 (25–710)	0 (0%)	2 (8.3%)	8 (6–76)

## Data Availability

Data available on request due to restrictions eg privacy or ethical.

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
