# Peer review of "Latest Findings on Minimally Invasive Anatomical Liver Resection"

_cancers, 2023, doi:10.3390/cancers15082218_

Round 1

Reviewer 1 Report

Dear Authors,   The manuscript entitled "Latest findings on minimally invasive anatomical liver resection" is an interesting technical description of minimally invasive anatomical hepatectomies from a specialized center headed by professor Go Wakabayashi.  The manuscript describes the frontier of minimally invasive liver surgery. However, it could be improved to achieve a higher standard.   Major points:   a) In manuscripts describing a surgical technique, it is crucial describing, at least, perioperative results, such as conversion, free margins, surgical complications (biliary leakage, bleeding, abdominal collection, etc.), and mortality. If possible, long-term results would reinforce the safety of the procedure. b) What are the diagnoses of patients who underwent MIALR? Is this technique applicable to all liver diseases? What are the indications for MIALR? c) Which liver surgery centers worldwide published papers about MIALR? It is interesting to discuss technical differences in MIALR. d) How do the authors think about the reproducibility of MIALR? It is important to discuss how that technique could be reproduced by other centers. e) The Introduction Section states: "This paper describes the latest findings on MIALR and our MIALR technique using ICG negative staining." The Conclusions Section states: "The Tokyo 2020 terminology is the first update on MILR since the publication of the Brisbane 2000 terminology and represents a major milestone in defining the guidelines for MIALR. In this article, we reviewed the highlights of the update." It is vital for the Conclusions to answer the aims of the study.    Minor points:    a) Line 48: MILAR must be replaced by MIALR b) Table 1: Please provide the meaning of HrS, Hr1, Hr2, and Hr3. c) The format of the References should follow Cancers authors guidelines, (e.g. Journal Articles: 1. Author 1, A.B.; Author 2, C.D. Title of the article. Abbreviated Journal Name Year, Volume, page range) d) How many robotic-assisted hepatectomies were performed since April 2022?        

Author Response

Major points:  

  1. In manuscripts describing a surgical technique, it is crucial describing, at least, perioperative results, such as conversion, free margins, surgical complications (biliary leakage, bleeding, abdominal collection, etc.), and mortality. If possible, long-term results would reinforce the safety of the procedure.

I appreciate your kind comment. I have added to the manuscript our short-term surgical results. Since we are still reviewing the surgical results of our technique, such as long-term result, we have included in this manuscript what we are able to report at this time.

  1. What are the diagnoses of patients who underwent MIALR? Is this technique applicable to all liver diseases? What are the indications for MIALR?

Thank you for your comment. We have added a new table on indications for our cases (Table1). For indication, I have additionally described the indications for MIALR on page2 line82.

  1. Which liver surgery centers worldwide published papers about MIALR? It is interesting to discuss technical differences in MIALR.

I would like to add to the discussion section (Paragraph 2) regarding reports from other institutions on MIALR. Especially from a technical aspect, I realized it was important to discuss the difference between positive and negative staining.

  1. How do the authors think about the reproducibility of MIALR? It is important to discuss how that technique could be reproduced by other centers.

We believe that the key word regarding reproducibility is “standardization of surgical procedure”. I would like to add to the discussion section (Paragraph 4) regarding this point.

  1. The Introduction Section states: "This paper describes the latest findings on MIALR and our MIALR technique using ICG negative staining." The Conclusions Section states: "The Tokyo 2020 terminology is the first update on MILR since the publication of the Brisbane 2000 terminology and represents a major milestone in defining the guidelines for MIALR. In this article, we reviewed the highlights of the update." It is vital for the Conclusions to answer the aims of the study. 

 I have revised the manuscript so that the Introduction and the Conclusion are consistent.

Minor points:   

  1. a) Line 48: MILAR must be replaced by MIALR

The misspelling was corrected. Thank you very much for letting me know.

  1. b) Table 1: Please provide the meaning of HrS, Hr1, Hr2, and Hr3.

I have appended explanatory text regarding these abbreviations.

  1. c) The format of the References should follow Cancers authors guidelines, (e.g. Journal Articles: 1. Author 1, A.B.; Author 2, C.D. Title of the article. Abbreviated Journal NameYearVolume, page range) .

I have corrected the format according to the guidelines.

  1. d) How many robotic-assisted hepatectomies were performed since April 2022?       

We have experienced 21 cases to date. This is additionally described in the manuscript.

Reviewer 2 Report

This manuscript was written by using the data including the 207 laparoscopic anatomical resections collected from a single center according to “The Tokyo 2020 terminology of liver anatomy and resections”

Criticisms

1.      What is the reason for excluding lateral segmental resections, as described in line 79 of page 2?

2.     In the table 1;

1)     What means the terminology ‘central bisegmentectomy’? Does it mean ‘central bisectionectomy’ or another concept? Then, it needs description and clarification. There is no description about this in “The Tokyo 2020 terminology of liver anatomy and resections”

2)     The numbers of HrS and HrSS are the same as 112. And it is not matched to 88 and 24 of the procedure in detail, segmentectomy and subsegmentectomy of the right column. Is it correct?  

3.     The concept of segmentectomy and subsegentectomy on the basis of staining by ICG fluorescent dye comes to readers as the description in the Tokyo 2020 terminology, whether it is the Glissonean pedicle approach or dye injection into the portal vein branch. Because the segmentectomy and subsegmentectomy occupy more than half of the entire cohort.

Reviewer is interested in which method the authors used for segmentectomy and subsegmentectomy. Both methods were used in this paper. What is the trial and success number and rate with each trial? Because the feeding portal vein(s) are not always large bored enough to apply these methods, frequently the branch(s) are too small or distributed. Therefore, an additive description of the number of each method with trial and success rate is needed.   

Author Response

1 What is the reason for excluding lateral segmental resections, as described in line 79 of page 2?

I appreciate your kind comment. The Japanese Society of Hepatobiliary and Pancreatic Surgery's definition of a high-difficulty liver resection does not include lateral segmental resections. However, in this manuscript, we have added six cases because we realized it was appropriate to include lateral segmental resections.

  1. In the table 1;

1)     What means the terminology ‘central bisegmentectomy’? Does it mean ‘central bisectionectomy’ or another concept? Then, it needs description and clarification. There is no description about this in “The Tokyo 2020 terminology of liver anatomy and resections”

I appreciate your kind comment. I have corrected my mistake and changed it to “bisectionectomy”.

2)     The numbers of HrS and HrSS are the same as 112. And it is not matched to 88 and 24 of the procedure in detail, segmentectomy and subsegmentectomy of the right column. Is it correct?  

Thank you for pointing out my mistake. I have corrected the mistake.

  1. The concept of segmentectomy and subsegentectomy on the basis of staining by ICG fluorescent dye comes to readers as the description in the Tokyo 2020 terminology, whether it is the Glissonean pedicle approach or dye injection into the portal vein branch. Because the segmentectomy and subsegmentectomy occupy more than half of the entire cohort.

Reviewer is interested in which method the authors used for segmentectomy and subsegmentectomy. Both methods were used in this paper. What is the trial and success number and rate with each trial? Because the feeding portal vein(s) are not always large bored enough to apply these methods, frequently the branch(s) are too small or distributed. Therefore, an additive description of the number of each method with trial and success rate is needed.   

I appreciate your comment. As you point out, sometimes it is difficult to identify the target Gleason during the Glissonean approach. However, with the use of an intraoperative 3D monitor, we are able to secure the target Glisson in almost all cases. We consider the use of intraoperative 3D monitor to be an essential device for the Glissonean approach and ICG NS.

Round 2

Reviewer 1 Report

Dear Authors,

The answers to the previously raised questions are satisfactory and the manuscript has improved significantly after corrections.